# Phylogeography of Organophosphate Resistant *ace* Alleles in Spanish Olive Fruit Fly Populations: A Mediterranean Perspective in the Global Change Context

**DOI:** 10.3390/insects11060396

**Published:** 2020-06-26

**Authors:** Esther Lantero, Beatriz Matallanas, Susana Pascual, M. Dolores Ochando, Carmen Callejas

**Affiliations:** 1Physiology and Microbiology, Department of Genetics, Faculty of Biological Sciences, University Complutense of Madrid, Av. José Antonio Novais 12, 28040 Madrid, Spain; estherlantero@ucm.es (E.L.); beatrizmp@bio.ucm.es (B.M.); dochando@ucm.es (M.D.O.); 2Department of Health Sciences, Faculty of Biomedical and Health Sciences, Universidad Europea de Madrid, c/Tajo, s/n, 28040 Madrid, Spain; 3Entomology Group, Department of Crop Protection, Instituto Nacional de Investigación y Tecnología Agraria y Alimentaria (INIA), Ctra de la Coruña km 7, 28040 Madrid, Spain; pascual@inia.es

**Keywords:** ***Bactrocera oleae*** (***olf***), insecticide resistance, organophosphates, ***ace*** gene, phylogeography, exon IV, exon VII, exon X ∆Q3

## Abstract

The olive fruit fly (***olf***) ***Bactrocera oleae*** is the most damaging olive pest. The intensive use of organophosphates (OPs) to control it, led to an increase in resistance in field populations. This study assesses the presence and distribution of three mutations at the ***ace*** gene related to target site insensitivity to OPs in Spain. Samples from other Mediterranean countries were included as external references. Resistance-conferring alleles (from exons IV and VII of the ***ace*** gene) reached almost an 80% frequency in ***olf*** Spanish populations. In total, 62% of them were homozygous (***RR/RR***), this being more common in eastern mainland Spain. High frequencies of ***RR/RR*** individuals were also found in North Mediterranean samples. Conversely, in Tunisia, only sensitive alleles were detected. Finally, the exon X mutation ∆Q3 had an extremely low frequency in all samples. The high frequency of genotype ***RR/RR*** in Spain indicates high fitness in an agroecosystem treated with pesticides, in contrast to ∆Q3. At exon IV all flies carried the same haplotype for the allele conferring resistance. The sequence analysis at this exon suggests a unique origin and fast expansion of the resistant allele. These results provide evidence that OPs appropriate use is needed and prompt the search for alternative methods for ***olf*** pest control.

## 1. Introduction

Spain is the world’s leading producer and exporter country of table olives and olive oil [1]. It concentrates more than 2.5 million ha of olive groves and 45% of the worldwide production [2,3]. This trade implies about EUR 1.7 billion of annual benefits [4].

The olive fruit fly (***olf***), ***Bactrocera oleae*** (Rossi, 1790), is the most damaging olive pest. Crop losses include reduced yield, low oil content, altered organoleptic properties and poor quality of olive products. Consequently, the ***B. oleae*** pest involves great economic costs in prevention and control of its populations.

Since the mid-20th century, this pest has been basically controlled by aerial insecticide spray treatments over large areas to reduce adult population peaks. The treatments have relied on synthetic insecticides, such as organophosphates (OPs) [5]. OPs inhibit acetylcholinesterase (AChE), encoded by the ***ace*** gene, blocking its active site, preventing the degradation of the neurotransmitter acetylcholine and resulting in the insect death by paralysis. The negative side effects of these compounds on the environment—mainly pollution and biodiversity loss—are of great concern [6,7]. Broad spectrum insecticides such as OPs are by far the most toxic compounds to non-target arthropods, becoming one of the main drivers of the entomofauna decline worldwide [8,9]. Such species decline is expected to lead to a steady decrease in insect-mediated ecosystem services. These ecosystem services are crucial for current intensive agriculture, where large monocultures, as olive groves, create the suitable conditions for pest development [10] Then, OPs overuse can lead to pest resurgence, due to the suppression of natural pest population control.

Because of the overuse of pesticides for decades, resistance to OPs in ***B. oleae*** populations has been reported since 1970 [11]. Vontas et al. [12] reported that species resistance was associated with an alteration in AChE, which results from point mutations located within the active site gorge of this enzyme. They compared cDNA sequences of individual olive flies from a susceptible strain with *olf* individuals from a resistant strain they obtained by laboratory selection, coupled with an enzyme inhibition assay by using omethoate. Their study revealed a glycine-serine substitution (G488S) at an amino acid residue, which is highly conserved across species and maintains the structure of the protein active center, as a likely cause of AChE insensitivity. This amino acid change was associated with a 35–40% reduction in AChE catalytic efficiency. The second one is a point mutation located in the exon IV that changes the amino acid valine for isoleucine next to the catalytic center (I214V) and reduces the insecticide’s ability to disable this site [12,13]. The substitution G488S, together with the substitution I214V (I199V substitution in ***Drosophila***, which confers low levels of OP resistance) produced up to a 16-fold decrease in insecticide sensitivity and, then, an important enhancement of insecticide resistance [12]. Finally, a deletion of three glutamines in the C-terminal of the protein (∆Q3) was described. This modification increases the affinity for the phosphatidil inositol present at the cell membranes and prolongs the presence of this enzyme in the synaptic gap [13,14,15]. Therefore, resistance of the olive fruit fly to OPs is related to three polymorphisms placed at three of the ten exons of ***ace*** gene, I214V at exon IV, G488S at exon VII and ∆Q3 at exon X [13].

The current knowledge on genetic resistance status for olive fruit flies field populations is only available for a few specific regions. ***Olf*** carrying resistant alleles to OPs has been observed in field populations from Greece, Italy, Cyprus and Turkey [16,17,18,19,20]. Indeed, the frequencies reported for IV–VII exons resistant alleles encompassed values between 69%–90% for Greek populations, between 75%–83% for Italian and about 68% for Turkish populations [16,18,20].

Recently, the Spanish Insecticide Resistance Action Committee (Spanish IRAC) warned of the risk of ***B. oleae*** resistance to OP insecticides, currently the main tools for its control [21]. Unfortunately, yet aside from a few characterized samples, the genetic study of Spanish ***olf*** populations regarding insecticide resistance is scarce, despite the agro-economic role of olive crops in this country. Only a few Andalusian olive flies (Southern Spain) have been sampled and results are highly variable; namely, in nine specimens the estimated average frequencies were near 33% for OP-resistant alleles and up to 80% for forty flies from four Andalusian populations, three of which were re-evaluated four years later showing similar values [16,22,23].

The increase in alleles conferring resistance to insecticides in field ***B. oleae*** populations may result in pest control failure. So, the monitoring of resistance levels is relevant for the management of this species. Considering all above, the central goal of this work was the phylogeographic study on genetic resistance status of ***B oleae*** populations in Spain. For this first Spanish comprehensive analysis, the three mutations of the ***ace*** gene were screened at twelve olive fruit fly populations across the entire olive growing area of Spain. Populations from other Mediterranean countries were also included to get an inclusive vision of the problem in a bid for tracking the origin of OPs resistance.

The European Union emphasizes that the use of pesticides ‘shall have the least effects on human health, non-target organisms and the environment’ [24]. Likewise, some studies point out that climate change is likely to affect insecticide resistance in the field [25]. Therefore, the information obtained in this survey might have a special significance for pest control in a more efficient, cost-effective and friendly environmental way in the worrying Global Change current frame.

## 2. Materials and Methods

### 2.1. Materials

***Bactrocera oleae*** from 12 sites across its complete distribution range in Spain were examined (Table 1). Additionally, samples from another 12 Mediterranean sites belonging to 5 countries were also included, containing, for the first time, two Tunisian samples. A total of 226 olive fruit flies collected from 24 olive growing areas were analyzed, on average 10 specimens/sample (Table 1). Infested olives were collected from Spanish olive groves, allowing larvae development in climatic cameras at the laboratory. The emergent adults were then stored at −20 °C. In abroad populations, adults were directly sent in ethanol (70%) and samples were conserved at 4 °C until the DNA extraction.

### 2.2. Methods 

Individual genomic DNA was extracted using the DNeasy Blood and Tissue Kit (Qiagen, Valencia, CA, USA) with some modifications. Integrity and concentration of the DNA was checked spectroforetically and electrophoretically on 0.8% agarose gels with TBE (40 mM Tris-Borate, 1 mM EDTA pH 8.0) stained with ethidium bromide (0.5 μL/mL).

#### 2.2.1. Exon IV

The primers BoAce-518F and BoAce-1040R were used to amplify a 521 bp fragment of the exon IV (543 bp in length) [26]. PCR reactions were carried out in a volume of 12.5 µL with 10 ng of genomic DNA, 1× buffer, 1.5 mM MgSO_4_, 200 µM of each dNTP, 0.65 µM of each primer and 0.2 U of the high fidelity Vent^®^ DNA polymerase (New England Biolabs, Ipswich, MA, USA). All PCR reactions were performed in a LabCycler apparatus (SensoQuest, Göttingen, Germany). The program included an initial denaturation step at 94 °C for 5 min, followed by 10 cycles of 94 °C for 30 s, 61 °C for 30 s and 72 °C for 30 s, and 20 cycles of 94 °C for 30 s, 57 °C for 30 s and 72 °C for 30 s plus a final extension step of 72 °C for 10 min.

Amplicons were checked on 1.5% agarose gels with TBE buffer and ethidium bromide. They were then purified with enzymes Exo I (Exonuclease I) and FastAP (Thermosensitive Alkaline Phosphatase), according to the instructions given by the manufacturer (Thermo Fisher Scientific, Waltham, MA, USA). Both strands were sequenced using the BIG Dye^®^ Terminator Cycle Sequencing Ready Reaction Kit (Applied Biosystems, Inc., California, CA, USA) in a 3730 DNA Analyzer (Applied Biosystems, Inc., USA) at the Genomics Unit of the Universidad Complutense of Madrid, Spain, using the same primers as in the amplification step.

Chromatograms were checked out; double peaks were recorded as heterozygous and subjected to cloning. These PCR products were cloned with the TOPO^®^ TA Cloning^®^ kit (Invitrogen, Carlsbad, CA, USA) using DH5α competent cells and following the instructions given by the manufactured. A minimum of 5 clones from each transformation were sequenced using the primers BoAce-518F and BoAce-1040R. All the sequences were aligned with CLUSTAL W software [27], edited with BioEdit v.7.0.9.0, and translated into amino acids to search for unexpected stop codons, ***indels*** and amino acid replacements [28].

#### 2.2.2. Exons VII and X

The primers pairs BoAce-1424/BoAce-1519 and Boace10F/Boace10R, were used to amplify exons VII and X respectively [16,18]. Amplifications were undertaken in a volume of 12.5 µL with 0.8 µM of each primer, 1.7 mM of MgCl_2_, 6.25 µL of Taq PCR Master Mix 2X (Qiagen) and 10 ng of DNA template. The PCR program included an initial denaturation step at 94 °C for 3 min, followed by 35 cycles at 94 °C for 30 s, 57 °C for 30 s and 72 °C for 30 s, and a final extension step at 72 °C for 3 min. Two microliters of PCR products were visualized in 2.5% agarose gels with TBE buffer and stained with ethidium bromide. The remainder amplification volume was incubated for 3 h with 7.5 U of enzyme ***BssHII*** (New England Biolabs, Ipswich, MA, USA) for exon VII or ***Mwol*** (New England Biolabs, Ipswich, MA, USA) for exon X at 50 °C or 60 °C, respectively, according to the instructions given by the manufacturer. The digestion products were electrophoresed in 4% agarose gels.

The allelic phase between exons IV and VII was assessed incorporating an extra base (A/G/R [IUPAC consensus base]) at the end of the exons’ IV alignment, representing the genotype obtained for the exon VII. Then, the new data set was analyzed with the algorithms implemented on DNAsp v4.50.3 [29].

## 3. Results

The 521 bp sequenced fragments of the exon IV of the ***ace*** gene revealed 10 polymorphic sites ***(SNP)***. Nine of them were silent mutations in the third codon position and the remaining, in the first position, caused the change from the aminoacid valine to isoleucine I214V (Table 2).

These 10 ***SNP***s defined 13 ***ace*** alleles, eleven sensitive to organophosphates (A1–A11) and two (A12 and A13), that confer OP resistance. The phylogeographic analysis showed the relationships among these 13 alleles, differing between 1 and 6 mutational steps (Figure 1). Alleles A12 (67.85%) and A1 (24.1%) were the most frequent, found in the 91% of the sampled flies. Both differ in five ***SNP***s. The resistant allele A12 was present in all Mediterranean countries except in Tunisia. A1, the most common sensitive allele, was found in all countries analyzed. The rest of the sensitive alleles detected (A2–A11) were identified at low frequencies. Variants A2 and A3 were characteristic of the Iberian Peninsula, Balearic Island and Tunisia. Variants A6 and A7 were detected in the Israeli and Tunisian populations. The A5 allele was present in flies from Israel, Tunisia and Greece. The rest of alleles—A4, A8, A9, A10, A11 and A13—were private from one country.

In Spain, three alleles for exon IV were detected, A12 (76.1%) and the sensitive to OPs A1 (23.47%) and A2 (0.43%). Genotyping revealed most of the individuals (63.5%) to be homozygous (***RR***) for the insecticide resistance ***SNP*** I214V. Some flies of the western Spain samples carried this mutation in heterozygosis (***RS***) (Figure 2). Conversely, Balearic flies (SPA5), revealed only sensitive genotypes in homozygosis (***SS***). The frequency of the point mutation G488S, located in exon VII and screened by PCR-RFLP (Appendix A), was very similar to that obtained in exon IV. Most of the Spanish olive fruit flies carried this ***SNP*** in homozygosis (***RR***), but the flies from Balearic population held insecticide-sensitive genotype to organophosphates (***SS***). By contrast, the 9 bp deletion in exon X, ∆Q3, also screened by PCR-RFLP (Appendix A), was found in only 2 of the 115 Spanish flies analyzed and always in heterozygosis. Upon examination of their genotypes for exons IV and VII, both flies were ***RS/RS*** (SPA9) and ***SS/SS*** (SPA5), (Appendix A).

A joint analysis of IV and VII loci showed that 62.25% of the Spanish olive flies held the I214V and G488S polymorphisms in homozygosis (genotype ***RR/RR***) (Appendix A). The second genotype in frequency was the double heterozygous (26.01%) with one chromosome carrying the sensitive alleles and the homologous the two OP resistant mutations, in coupling phase, with a 95% of probability (***RS/RS***). Just 11.74% of the Spanish fly samples were double homozygous for the sensitive alleles (excluding the insular sample, the frequencies were 68.6%, 27.6 and 2.8%, respectively, for ***RR/RR***, ***RS/RS*** and ***SS/SS*** genotypes at Spanish mainland ***olf***).

Figure 2 provides an informative phylogeographic picture of these outcomes in Spain. In short, no genetic diversity was found at three populations (SPA3, SPA10 and SPA11), with a 100% homozygous genotype (***RR/RR***) for the two-point mutations. Only two out of the 11 peninsular samples (SPA1 and SPA4) held some olive fruit flies with genotypes insecticide-sensitive to organophosphates (***SS/SS***). Figure 2 also shows the distribution pattern of ***ace*** gene alleles in ***olf*** Spanish populations. Western samples had a higher frequency of sensitive alleles in homozygosis and heterozygosis, also observed in the Portuguese ones. The proportion of alleles with ***SNP*** for OP resistance increased in flies from Central Spain, with higher heterozygous and homozygous resistant-insecticide genotypes, whereas the eastern Spanish ***olf*** populations had a higher frequency of these alleles conferring OP resistance, and generally in homozygosis. As stated above, Balearic flies revealed only sensitive genotypes in homozygosis.

The analysis of the three described mutations at ***ace*** gene (I214V, G488S and ∆Q3) also yielded informative results for the other five Mediterranean countries sampled. On one hand, the Italian and Greek flies had no OP sensitive alleles in homozygosis for exons IV and VII; their joint analysis scored the highest values of double homozygotes for OP resistance alleles ***RR/RR*** (Figure 2 and Appendix A). Conversely, the Tunisian populations sampled showed a significant genetic diversity for sensitive alleles, seven scored (Figure 1), but no ***RR*** homozygotes for exons IV and VII of ***ace*** gene were detected. Somewhere in between these two extremes was Israel, which also attained a high genetic diversity with seven sensitive alleles at exon IV, three of which were private (A9–A11, Figure 1). Near 50% of the ***olf*** surveyed from Israeli olive fruit flies carried in heterozygosis an OP-resistant allele in exons IV and VII (***RS***), and nearly a third (27.6%) of flies studied were double homozygous (***RR/RR***) for both ***SNP***s. Finally, in Portuguese samples, the double homozygous frequency for the resistant alleles (genotype ***RR/RR***) decreased by up to 7.6% (Figure 1).

As in Spanish samples, the deletion in exon X was found at very low frequencies, and always in heterozygosis, in the Israeli, GRE1 and POR1 samples. In particular, the three Israeli samples presented it as 20.7% on average [2 flies had SS/SS genotypes for exons IV and VII, 2 were ***SS/RS*** and 2 showed other combinations (Figure 2 and Appendix A), (Appendix A)]. ∆Q3 was also found in 2 flies from the GRE1 sample (whose genotypes for exons IV and VII where ***RR/RR*** and ***RS/RS***), and in one specimen from the POR1 sample (genotype ***RR/RR*** for the two-point mutations).

In summary, the Balearic was the only European sample that did not show ***SNP*** for resistance to OPs. The highest frequencies of both insecticide-resistant polymorphisms (alleles) were in populations from Spain (0.76 and 0.74; in mainland Spain 0.83 and 0.81), Italy (0.95 and 0.95) and Greece (0.97 and 0.95), mostly in homozygosis for both loci, respectively. The Israeli and Portuguese samples showed a frequency of resistant alleles near to 50%, mainly in heterozygosis. It should be noted the minute or null frequency of the alleles that confer resistance to the organophosphates in Tunisian samples. Figure 2 also displays the frequency and distribution of point mutations in exons IV and VII of the ***ace*** gene in these Mediterranean olive fruit fly populations.

## 4. Discussion

Four interesting findings can be highlighted in this work. Firstly, the Spanish olive fruit fly populations as a whole have a high frequency of ***SNP*** conferring resistance to organophosphates, both in IV and VII exons. Most of the flies, 62.25%, were double homozygous (genotype ***RR/RR***) for the point mutations (Figure 2). This genotype was found in all Spanish peninsular samples, regardless of the olive grove management regimes. Previous laboratory studies indicate that the combined presence of both polymorphisms (in exons IV and VII) does not impair fly survival, even without exposure to insecticides [12]. Indeed, the evaluation in vitro of the acetylcholinesterase activity of olive fruit fly lab strains showed an enzyme 16 times more insensitive to organophosphates in those flies carrying the alleles that confer resistance in homozygosis in exons IV and VII than in flies with sensitive alleles [30]. These individuals maintained an 80% catalytic activity, whereas in those olive flies homozygous only for the OP-resistance SNP in exon VII, the activity decreased to 40%. Such results led Hawkes et al. to suggest that the presence of the mutation in exon IV would reduce the adverse effects caused by the mutation in exon VII in acetylcholine catalysis [16]. Consequently, the occurrence and great frequency of genotype ***RR/RR*** in Spanish olive groves would reflect the selective advantage of olive flies carrying the double homozygote insecticide-resistant genotype in an environment treated with OPs. Likewise, ***olf*** carrying resistance alleles appear to survive without great biological cost when the grove is no longer treated with insecticides, or in untreated olive groves.

Considering the high dispersal capacity of olive flies, gene flow also seems to have influenced the frequency and wide distribution of these two polymorphisms in the Spanish populations of ***B. oleae***. This result is especially relevant in terms of pest management, since the vast connection between populations would favor the rapid dispersion of genetic variants, including those related to insecticide resistance. The huge land cover of olive groves in Spain has been proposed as one of the significant causes for the unrestricted gene flow among Spanish mainland ***olf*** populations that, in fact, can be considered as a single large population [31,32].

The phylogeographic analysis also revealed the distribution pattern of ***ace*** gene alleles in Spain (Figure 2). Strikingly, the olives flies from Balearic Islands (SPA5) carried only sensitive alleles in both exons in homozygosity. This could be a consequence of a possible drift effect on the olive fly populations given the particularity of the insular territory of Mallorca. In contrast, ***ace*** OP-resistance alleles were detected in Crete and Cyprus samples [18,20], both islands are also very close to the mainland, albeit larger than Mallorca. Samples SPA3, SPA9 or SPA12 showed a higher frequency of OP-resistance alleles. Towards the west, at populations as SPA1, SPA4 or SPA6, the proportion of flies with one (***RS***) or two copies (***SS***) of DNA in IV and VII exons increases. Portuguese samples also presented a higher frequency of sensitive alleles in homozygosis and heterozygosis (Figure 2).

This trend may be partly due to different selective pressure. In coastal regions, where temperatures are milder and the humidity higher, more generations of olive fruit flies must occur than in the inner peninsula, as the number of annual generations of ***B. oleae*** is dependent on temperature and humidity [33]. This fact in turn translates into differences in the intensity of the pest attacks and the different need to apply control measures, which would be greater in coastal areas. Likewise, an increase in the efficacy of selection due to more bouts of selection and an increased effective population size might explain such trend. It is also necessary to consider the type of olive grove management in each region. Since 1992, the Community of Extremadura (western Spain) has led the national organic agricultural production (Junta de Extremadura, Directorate-General for Agriculture and Livestock), of which approximately 35% is dedicated to olive groves (data from 2016). This entails a fairly limited use of chemical treatments against pests (General Sub-Directorate of Differentiated Quality and Ecological Agriculture, MAPAMA). Then, there would be less selection pressure on ***B. oleae*** populations in this area, which could explain, to a certain extent, the lower frequency of resistance to organophosphate insecticides detected (SPA4 and SPA6). Similar or lower frequencies to ours have been estimated from Portuguese samples in other works. A recent report [23] revisited the study of Pereira-Castro et al. about ***ace*** alleles in eight Portuguese sites (80 flies) [22]. The most frequent genotype found was ***RS/RS*** (49%) although they observed an increase in ***ace*** gene point mutations compared to their previous survey. The low selective pressure exerted over decades due to a reduced number of insecticide treatments in these olive groves may have played a relevant role in the low level of resistance [34]. Therefore, it is possible that the frequency of OP-resistant alleles in Portugal will increase due to the current, more intensive olive grove production, and to the westward dispersion of olive flies carrying OP-resistant alleles given the unrestricted flow [32].

Focusing on the other Mediterranean samples analyzed, their number and their sample size are similar to, or sometimes greater than, those attained in other works, allowing the comparison and monitoring of ***B. oleae*** populations [16,17,18,19,20]. Figure 2 and Appendix A show the pervasive frequency and distribution of double homozygous flies for the two point mutations along the North Mediterranean area. The elevated values of RR/RR genotypes found at Italian and Greek samples (90% each) are akin to previous estimates [16,18,19], evidencing the need of the rational use of insecticides in the field. The strong association between I214V and G488S polymorphisms in the ***B. oleae**ace*** gene was also observed in other works and in other species as well (***D. melanogaster***’ and ***B. dorsalis***) [12,18,19,22,23,35,36]. Most likely, a recombination event would link these two ***SNP***s acquired independently at DNA exons IV and VII of the gene instead of the appearance of a new point mutation in ***ace*** gene that already carries a polymorphism that confers resistance [35].

Secondly, and also outstanding, was the detection for the first time of the two-point mutations in the ***ace*** gene related to the OPs insensitivity in all the ***olf*** Israeli samples. No ***ace*** OP-resistance alleles were found in this country in the only study available in the literature [26]. The present analysis revealed 50% genetic resistance, mostly as heterozygosis (Figure 2), and a great diversity in the number of sensitive alleles at exon IV—seven, three of which were private. Such diversity of ancestral and derived alleles may be indicative of long-established and large ***B. oleae*** populations. Monitoring of ***olf*** populations will be interesting, because the Israeli government is limiting the use of agricultural chemicals [37], which would reduce the selection pressure on these populations.

Thirdly, only sensitive genotypes have been found at Tunisian samples, never analyzed before. As in Israel, a great diversity in the number of sensitive alleles at exon IV was found—seven, one being private. The standing variation and sharing of ancestral sensitive alleles with the Israeli populations seems to be borne out of the long stablished and large ***B. oleae*** populations at this region. In addition, no alleles for OP resistance have been found in South African or Kenyan studies [16,26]. A similar result was reported on ***Ceratitis capitata***, where only one in 27 individuals was heterozygous for the resistant allele of the ***ace2*** gene, despite the current use of OP for pest control [38,39]. Future surveys would therefore be advisable to monitor how frequencies evolve.

Fourthly, another remarkable finding is the 9 base pairs deletion of exon X in two Spanish samples (SPA5 and SPA9), also in the Iberian POR1, but at a very low frequency (near 2.5%), and always in heterozygosis. Kakani et al. analyzed 60 specimens from two Andalusian sites screening this deletion [18]. They did not detect it in Spanish samples, but they did in the Portuguese one (from Lisbon). Nobre et al. did not find this deletion in the Iberian samples analyzed [23]. As in Spain, ∆Q3 frequencies in the east-Mediterranean basin were very low, the estimation for Greek populations (6.89%) agrees with a previous report (6.45%) [18]. Furthermore, as a novelty, deletion was found at the three ISR samples too. This generalized low and patched frequency points to a low selective value of olive flies carrying this mutation in exon X, even in heterozygosis and in an OP-treated environment. The combined analysis of the three mutations at once (I214V, G488S and ∆Q3), revealed that exon X deletion is usually associated with sensitive genotypes at exons IV and VII (genotype ***SS/SS***), and to a lesser extent to other combinations. These rare genotypes would support an even lower fitness in a pesticide-use frame of the olive flies carrying deletion ∆Q3 at exon X together with the ***SNP***s in the other two exons.

The occurrence of simple or multiple origins and mutations that confer resistance to insecticides is a subject of debate [40]. DNA sequencing can be successfully brought to bear on this long-standing question in population genetics. Single mutations to arise recurrently in different olive fly populations would be expected because of their big effective population sizes and their long time of being established in the Mediterranean. The DNA sequences yielded for exon IV were identical in all the olive flies carrying the polymorphism that causes the non-synonymous change in the protein and confers insecticide resistance, ***I214V***, except one (Table 2, Figure 1). This reduced or null genetic diversity in the SNP surrounding genomic region amongst individuals is potentially indicative of the recent positive selection of this ***SNP*** in an intense insecticide-use frame. Therefore, the marked haplotypic structure found suggests a soft selective sweep. The nucleotide sequence (allele A12) agrees with the ***ace*** allele OP resistant loaded in Genbank (accession number DQ499479) [26]. Then, the hypothesis of the unique origin of this mutation was borne out by the detection of a single allele at the exon IV ***ace*** gene that confers insecticide resistance to OPs in north- and east-Mediterranean olive fly populations. The phylogeographic analysis also showed that the A12 allele derives from the sensitive one, A5; both differing in a single ***SNP*** (Figure 1). The A5 allele is typically from Tunisia, Greece and Israel populations (Figure 1). Our results would support the possible geographic origin of the OP-resistance allele in the eastern Mediterranean regions, as suggested by other authors [16,18,26]. This allele was most likely part of the genetic pool of the olive fruit fly before the use of OP insecticides. Mutations associated with resistance that predate the use of insecticides have been reported in other flies, e.g., ***Lucilia cuprina*** [41], whose individuals carrying these variants would not have a relevant cost in the absence of insecticides [40,42]. Given the systematic and severe insecticide treatments in these Mediterranean areas, this advantageous polymorphism would be positively selected, increasing its frequency to its current almost fixation in most Greek and Italian ***olf*** populations (Figure 2 and Appendix A). Then it would also spread to the other north west-Mediterranean ***B. oleae*** populations because of the considerable gene flow detected among them [31,32]. This dispersal would be favored by the migratory capacities of the olive fruit fly, but also by the olive trade, as discussed elsewhere [20].

Therefore, the selective advantage that these DNA polymorphisms confer to the olive flies in an insecticide-treated environment, together with the high dispersion capacity of this species, would shape the levels and distribution patterns of the ***ace*** alleles found in the present work.

This information is useful for the productive sector. Currently, all the programs to control such pests include the use of chemical products; but with a longer growing season and a warmer climate, insect pests are probably on the rise, driving a greater use of pesticides. However, the increase in resistance to OPs and pyrethroids [43], the other major insecticide currently used against this pest in Spain, the negative side effects of pesticides on biodiversity and also on global change provide evidence that their appropriate use is needed and prompt the search for alternative methods.

## 5. Conclusions

Spanish olive fruit fly populations have a high frequency of genetic resistance to organophosphate insecticides, mostly as double homozygous (***RR/RR***) genotypes in the ***ace*** gene. These resistance alleles are widely distributed, which constitutes a challenge for the olive growing sector, not only in Spain, but also in the north of the Mediterranean basin. The moderate frequencies of Israeli samples and minute frequencies of the Tunisian samples provide good opportunities for follow-up studies to further the analysis the molecular basis of the origin and dispersal of resistance genes to insecticides in ***olf*** populations. As the number of products authorized for the control of B. ***oleae*** decreases, it is essential to safeguard their long-term effectiveness and their appropriate use. In order to reduce olive fly populations and prevent resistance, it is necessary to promote the rational use of insecticides, together with other available control alternatives, such as cultural, biological, mass trapping, attraction and death techniques, etc., as recommended by the IRAC [21].

## Figures and Tables

**Figure 1 insects-11-00396-f001:**
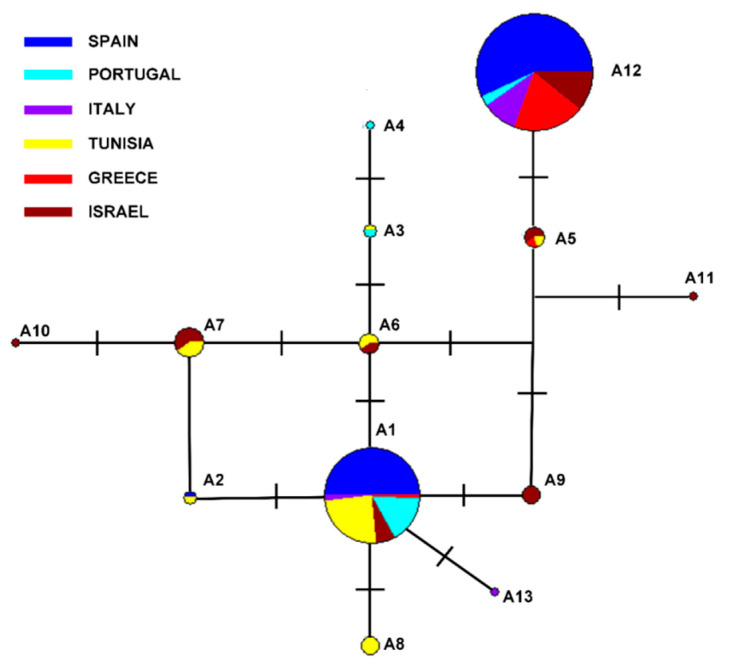
Median Joining Network of the 13 alleles found in the Mediterranean ***B. oleae*** populations for exon IV of ***ace*** gene. Alleles are represented by diagrams whose sizes are proportional to the number of individuals that carry them. The 13 alleles are separated by a minimum of 1 ***SNP*** and a maximum of 6. Colors represent the sampled countries.

**Figure 2 insects-11-00396-f002:**
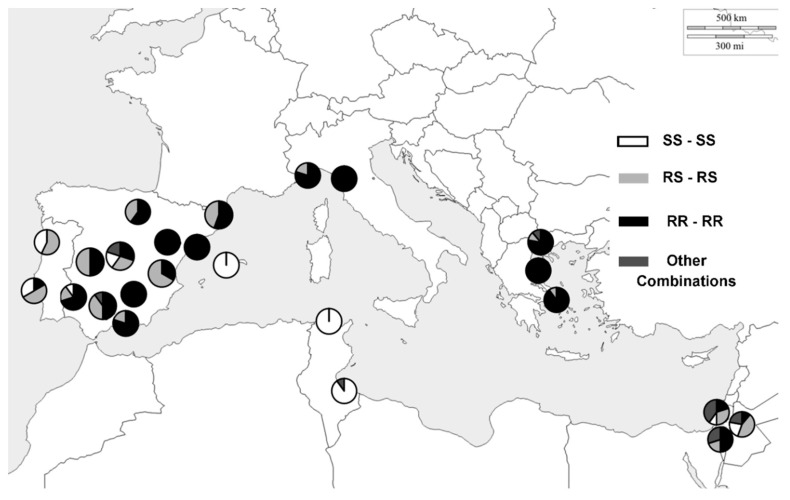
Genotype frequencies and distribution for the ***ace*** gene exons IV and VII in the populations of olive fruit fly sampled. Sector size is proportional to the number of double homozygotes or heterozygotes for both exons.

**Table 1 insects-11-00396-t001:** Collection data of the studied populations. LAT, latitude; LONG, longitude; N, number of individuals analyzed by population. SPA: Spain, POR: Portugal, ITA: Italy, TUN: Tunisia, GRE: Greece, ISR: Israel.

CODE	LOCALITY/COUNTRY	LAT	LONG	N
SPA1	Morata de Tajuña, Madrid, ES	40.2275	−3.4369	10
SPA2	Arróniz, Navarra, ES	42.4222	−2.0913	11
SPA3	Tortosa, Tarragona, ES	40.811	0.5209	10
SPA4	Montemolín, Badajoz, ES	38.1552	−6.2069	10
SPA5	Mallorca, Baleares Isles, ES	39.6952	3.0175	10
SPA6	Castañar de Ibor, Cáceres, ES	39.6277	−5.4166	10
SPA7	Campus Rabanales, Córdoba, ES	37.2647	−4.6327	10
SPA8	El Cortalet, Gerona, ES	42.2253	3.0970	10
SPA9	Íllora, Granada, ES	37.3461	−3.8727	9
SPA10	La Iruela, Jaén, ES	37.9469	−2.9583	10
SPA11	La Portellada, Teruel, ES	40.89	−0.0336	9
SPA12	Requena, Valencia, ES	39.4878	−1.1003	6
POR1	Fundao, Castelo Branco, PT	40.1369	−7.4994	7
POR2	Lisbon, Lisbon, PT	38.7069	−9.1356	6
ITA1	Diana Marina, Liguria, IT	43.9098	8.0818	10
ITA2	Pisa, Toscana, IT	43.7498	10.5497	10
TUN1	Sidi Thabet, Ariana, TN	36.9081	10.0222	10
TUN3	Zarzis, Médenine, TN	33.523	11.0852	10
GRE1	Agiá, Tesalia, GR	39.7188	22.7550	10
GRE2	Salónica, Tesalónica, GR	40.6393	22.9446	9
GRE3	Atenas, Central Atenas, GR	37.9791	23.7166	10
ISR1	Jerusalem, Jerusalem District, IL	31.7383	35.2137	9
ISR2	Rehovot, Central District, IL	31.8927	34.8112	10
ISR3	Lahav Forest, Southern District, IL	31.3725	34.8408	10

**Table 2 insects-11-00396-t002:** Alleles found and their frequency for exon IV ace gene in the 226 Mediterranean *B. oleae* analyzed. Grey shades mark the two most frequent alleles in the sample. In bold, the polymorphism that causes the non-synonymous change in the protein and confers insecticide resistance I214V. The asterisk indicates the same nucleotide as the A1 haplotype in that position of the sequence.

	73	88	122	259	274	316	406	412	415	484	Freq
**A1**	T	T	A	G	A	C	G	G	G	T	24.1%
**A2**	*	*	*	*	*	T	*	*	*	*	0.44%
**A3**	*	*	*	*	*	*	*	*	T	C	0.44%
**A4**	*	*	*	*	C	*	*	*	T	C	0.22%
**A5**	C	*	*	C	*	*	*	A	T	*	1.11%
**A6**	*	*	*	*	*	*	*	*	T	*	1.11%
**A7**	*	*	*	*	*	T	*	*	T	*	2.23%
**A8**	*	G	*	*	*	*	*	*	*	*	0.88%
**A9**	C	*	*	*	*	*	*	*	*	*	0.88%
**A10**	*	*	*	T	*	T	*	*	T	*	0.22%
**A11**	C	*	*	C	*	*	A	*	T	*	0.22%
**A12**	C	*	**G**	C	*	*	*	A	T	*	67.85%
**A13**	*	*	**G**	*	*	*	*	*	*	*	0.22%

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
