# Peer review of "Phylogeography of Organophosphate Resistant ace Alleles in Spanish Olive Fruit Fly Populations: A Mediterranean Perspective in the Global Change Context"

_insects, 2020, doi:10.3390/insects11060396_

Round 1

Reviewer 1 Report

The manuscript “Phylogeography of organophosphate resistant ace alleles in Spanish olive fruit fly populations. A Mediterranean perspective in the global change context” uses targeted sequencing of three known insecticide resistance loci to perform genetic resistance monitoring of the major olive pest Bactrocera oleaeThey find that resistant alleles in two of the three loci segregate at considerable frequencies in the Mediterranean, and further find geographic variation in their frequency – notably, whereas resistant alleles are rare in Portugal and nearly nonexistent in North Africa, they are common throughout Spain, Italy, Greece. Moreover, the limited genetic variation associated with resistant alleles leads to a hypothesis of recent strong selection for these alleles. I like the paper. Genetic resistance monitoring is a worthy endeavor, and the paper is straightforward and easy to read. The sample sizes for each location are a bit lower than I would like to see, but my confidence in the results is bolstered by the consistency between nearby sampling sites as well as consistency with other findings in the literature. I have some minor comments regarding the presentation of results and some of the introduction/discussion that I would like to see addressed before publication. 

Main comments: 

Introduction: 

Is the pattern of dominance for any of these mutations known? Also, does each mutation on its own give complete resistance, or are some more effective than others? Are the mutations redundant in their effects? There is a little bit of relevant information in the first paragraph of the discussion, but I would like to see more detail, and I think it would be appropriate in the introduction. Broadly, giving more detailed background on the biology underlying the association between the ace gene and resistance will allow the reader greater ability to interpret the results. 

Materials and methods: 

Please describe the method used for collecting the flies and storing them prior to DNA extraction. 

Figures/Tables: 

Table 2; Figure 1: For the sake of consistency and transparency, I would like to see versions of these for exons VII and X in addition to exon IV.  

Figure 1: The fact that there are so many more samples from Spain than other countries makes this figure slightly misleading. For example, it looks at first glance like haplotype A1 is extremely common in Spain, although the text indicates that it is actually only present in <25% of individuals. Perhaps the authors could weight the size of each pie slice by the total number of individuals from each country, which would give us a better idea of the frequency of each haplotype in each country rather than the raw number of individual flies with each haplotype. 

Figure 2: I would consider removing this figure. It really doesn’t present any meaningful information that isn’t present in Figure 3, which is much stronger and more interesting to look at. In my opinion, Figure 2 is more likely to confuse readers than to inform them. 

Specific comments: 

Lines 85-87: The end of this sentence is confusingly written and should be reworded. 

Line 119: Should read “same primers as in the amplification step” 

Lines 263-268: Another hypothesis for why regions with more generations of olf per year might contain more resistance alleles could be simply an increase in the efficacy of selection due to more bouts of selection and increased effective population size. 

Line 307: Remove the first “have” in “no have alleles for OP resistance have been found” 

Lines 328-331: This statement is circular – the A12 allele was defined as a mutlilocus haplotype, so of course all individuals with that allele were identical. Moreover, I think this statement as well as the following statement regarding linkage disequilibrium are irrelevant to the authors’ argument. The more important point, as they discuss from lines 332 onwards, is that the reduced haplotypic variation amongst resistant individuals is potentially indicative of recent selection for resistant alleles, and suggests a soft selective sweep.  

Generally, in this paragraph, the authors should be careful to frame their arguments as hypotheses rather than definitive findings, given that their results are descriptive and contain no formal population genetic analysis. 

Lines 354-358: This paragraph is vague and contains a number of grammatical errors, I would consider revising. 

Line 365: Replace “dispersion” with “dispersal” 

Reviewer 2 Report

An interesting and useful study.

Some changes to the manuscript are recommended.

Abstract:

Lines 19-20: change to '...related to target site insensitivity...'

Lines 29-30: The final sentence requires clarification - while widespread resistance to OPs is certainly a reason to reduce or stop using OPs, changing to more environmentally-friendly approach is primarily justified by the toxicity and environmental side effects of OPs.

Introduction:

Line 42: change to '...aerial insecticide spray treatments...'

Line 43: change to '...treatments have relied...'

Line 44: change to 'OPs inhibit acetylcholinesterases (AChE),

encoded by...'

Line 45: change to '...preventing the degradation of the neurotransmitter acetylcholine and resulting...'

Lines 46-50: rewrite to describe more accurately the negative side effects of insecticides; it is not true to say that insecticides as a group are by far the most toxic compounds to all insects and other arthropods..some are broad spectrum compounds such as OPS, which can be very harmful to non-target athropods including beneficial insects to humans (e.g. polinators, biocontrol agents), some other insecticide groups are more selective and have fewer side effects, e.g. insect growth regulators, macroclyclic lactones and products based on Bacillus thuringiensis toxins. A key point to make in the context of pest management is that broad spectrum insecticides such as OPs are often highly detrimental to pest natural enemies (parasitoids and predators) and their overuse can lead to pest resurgence due to suppression of natural pest population control.

Lines 56: change to 'insecticide's..' also, add ref 13 at end of sentence

Lines 58-59: this sentence not make it clear that the mutation in exon VII again leads to target site (AChE) insensitivity to OPs.

Lines 61-62: explain in the text why this impairs the effect of OPs and thus results in resistance.

Line 69: while pyrethroids are also used for Olive Fly control, mentioning them here rather than in the Discussion could confuse the reader as the present study was on altered target site resistance mechanisms that do not relate to these compounds. Suggest say '...to OP insecticides, currently one of the main tools...'

Lines 83-85: the first two sentences would be better as part of the Discussion; suggest start paragraph with 'The information obtained in this survey...'

Materials:

Lines 91-92: suggest change to 'Bactrocera oleae from 12 sites across its complete distribution in Spain were examined (Table 1).

Line 93: no need for a new paragraph.

Line 94: change to 'A total of 226...'

Methods:

Line 120: suggest change to 'Chromatograms were examined by eye...'

Results:

Line 173: suggest change to 'Examination of their geneotypes for exons....'

Line 195: change to 'As stated above, Balearic...'

Discussion:

Since four interesting findings are mentioned in Line 230, it would be helpful to the reader to signpost the second, third and fourth findings, e.g.           Line 247: 'Secondly, considering the high dispersal...'                              Line 295: 'Thirdly, this is first time ....'                                                     Line 311: 'Fourthly, ...'

Line 257: 'In contrast...' instead of ..'Contrarily...'

Line 325: change to '..is a subject of debate.....

Lines 354-358: the final paragraph would benefit from greater clarity [e.g. bring together longer growing season/warmer climate (Lines 355-356), with climate change (Line 357)] and expansion, e.g. this might be the place to refer to pyrethroids - the other major insecticides currently used against this pest in Spain - which also suffer from the same issues of resistance and environmental side effects [Refs].

Conclusions:

Line 367: is it a good idea to safeguard the long term effectiveness of OPs when they are environmentally harmful?
